# Dehydration Impairs Physical Growth and Cognitive Development in Young Mice

**DOI:** 10.3390/nu12030670

**Published:** 2020-02-29

**Authors:** Chong-Su Kim, Woo Young Chun, Dong-Mi Shin

**Affiliations:** 1Department of Food and Nutrition, College of Human Ecology, Seoul National University, Seoul 08826, Korea; jskim11@snu.ac.kr; 2Department of Psychology, Chungnam National University, Daejeon 34134, Korea; 3Research Institution of Human Ecology, Seoul National University, Seoul 08826, Korea

**Keywords:** dehydration, physical growth, cognitive development, brain transcriptome, hippocampus, brain-derived neurotrophic factor

## Abstract

Infancy and childhood are periods of physical and cognitive development that are vulnerable to disruption by dehydration; however, the effects of dehydration on cognitive development during the periods have not yet been fully elucidated. Thus, the present study used a murine model to examine the effects of sustained dehydration on physical growth and cognitive development. Three-week-old C57BL/6J mice were provided either ad libitum (control group) or time-limited (15 min/day; dehydration group) access to water for 4 weeks. Physical growth was examined via a dual-energy X-ray absorptiometry whole-body scan, and cognitive development was assessed using the Barnes maze test. RNA-sequencing and qPCR analyses were carried out to assess the hippocampal transcriptome and the expression of key neurotrophic factors, respectively. These analyses showed that dehydrated mice exhibited a reduced body mass and tail length, and they spent four times longer completing the Barnes maze test than control mice. Moreover, dehydration significantly dysregulated long-term potentiation signaling and specifically decreased hippocampal brain-derived neurotrophic factor *(Bdnf)* expression. Collectively, these data confirm dehydration inhibits physical growth and suggest that it impairs cognitive development by altering the hippocampal transcriptional network in young mice; thus, they highlight the importance of water as a vital nutrient for optimal growth and development during infancy and childhood.

## 1. Introduction

Water in the body acts as a solvent to carry nutrients, a reactant and product during metabolism, and a major constituent of both cells and tissues [1]; thus, optimal hydration is essential to enable a range of normal physiological functions required to maintain organismal health [2]. While compelling evidence supports that dehydration negatively affects physical and physiological functions [3,4,5,6], its effects on cognitive function remain controversial. For example, some previous studies have reported that fluid restriction negatively impacts short-term memory [7], while conversely, others have shown that mild dehydration improves short-term memory function [8]. Moreover, although the importance of water as an essential nutrient is well established, the mechanisms underlying the negative impacts of dehydration, especially on cognitive function, have not yet been fully elucidated. Therefore, finely controlled animal studies are urgently needed to further examine how inadequate daily water intake detrimentally impacts cognitive function.

Water requirements to achieve adequate hydration differ with age [9]. Children require a higher water intake/unit of body weight than adults because they have a higher body-water composition ratio [10], and they are at a greater risk of developing dehydration due to their immature fluid regulatory systems [11]. Moreover, childhood encompasses critical periods of physical and brain development that require adequate water intake to facilitate normal physical and neurodevelopmental processes [12]. Therefore, infants and children are more vulnerable to dehydration than adults. 

Cognitive development, including particularly learning and memory formation, relies heavily on hippocampal long-term potentiation (LTP) [13]. Brain molecules such as neurotrophins regulate diverse processes including neuronal development, synaptogenesis, and plasticity [14,15,16]. Among these, brain-derived neurotrophic factor (BDNF) plays a critical role in cognitive plasticity during learning and memory formation [13,15,17]. A growing body of evidence indicates that reducing BDNF levels decreases the number of maturing neurons [18], reduces neurotransmission, and alters LTP in the hippocampus [15,16,19]. Given its critical roles in neuronal development, synaptogenesis, and synaptic plasticity, we postulated that BDNF may mediate the effects of dehydration on cognitive function.

Thus, the present study sought to examine the effects of sustained dehydration during growth period on cognitive development at both the organismal and molecular level using a murine model. The findings of this study provide the first molecular evidence showing the importance of adequate hydration during infancy and childhood for normal physical and brain development.

## 2. Materials and Methods 

### 2.1. Experimental Animals 

Immediately after weaning, 3-week-old male C57BL/6J mice (Jackson Laboratory, Bar Harbor, Maine, USA) were randomly assigned into control (CON, *n* = 6) and dehydration groups (DEH, *n* = 7), group-housed (3–4 mice/cage), kept at a constant temperature (22 ± 2 °C) on a 12-h light/dark cycle (light from 08:00 am), and provided with ad libitum access to an AIN-93G diet. Body weight and water intake were recorded daily, both before and after provision of a water supply. The daily energy intake was calculated by multiplying the dietary intake (g/day) by the energy density (kcal/g) of the provided diet. After 4 weeks, mice underwent a 12-h fast before being euthanized via intraperitoneal injection of 20% urethane (U2500, Sigma-Aldrich, MO, USA) for immediate collection of blood and tissue samples. All experimental procedures were approved by the Institutional Animal Care and Use Committee (IACUC) of Seoul National University (SNU-140421-3) and were conducted in strict accordance with IACUC guidelines.

### 2.2. Water Restriction

Sustained dehydration was induced for 4 weeks from weaning as previously described [11]. Briefly, animals in the DEH group were provided access to a water supply for 15 min at the same time each day during the experimental period and after the completion of conducted behavioral assessments. CON mice were provided with ad libitum access to a water supply.

### 2.3. Behavioral Assessment

Cognitive function was assessed using the Barnes maze test, which is a standardized method for evaluating spatial learning ability in young rodents [20]. During the final week of water restriction, a behavioral assessment was conducted at the same time each day during the light phase (between 09:00–13:00) in a dedicated room. The utilized maze comprised an elevated (105 cm above the floor) circular platform (92 cm diameter) with 20 holes at the edge. One of these holes was connected to a dark tunnel that was linked to a hidden escape cage. Visual cues (i.e., different colors and shapes) were provided at the same points during the experimental period using a 30 W LED light that was installed above the platform.

Specifically, mice were subjected to the following:An adaptation phase—on the first day, animals were allowed to become accustomed to the experimental environment and task. They were placed into a cylindrical transparent start chamber in the middle of the maze, and after 10 s, were guided to the escape hole and allowed 3 min to freely enter the escape cage through the escape hole;A learning phase—next, mice were subjected to four consecutive days of training (comprised of two training trials/day, with a 15 min interval) to assess their spatial reference learning ability. Animals were placed into an opaque start chamber for 10 s and allowed to explore the maze for 3 min. If a mouse failed to find the escape hole within this time, it was guided to the hole.A test trial—on the final test day, mice were allowed only a single trial to find the escape hole.

All behavioral assessment trials were video recorded and analyzed using Ethovision XT 10 software (Noldus, Wageningen, Netherlands). Spatial learning ability was assessed by calculating the primary escape latency (s) and cumulative path length (cm) during training-phase and test trials. The speed (cm/s) was calculated by dividing the total distance moved by the primary latency during the 4-day training phase and test trials. The relative latency (%) required to identify the escape hole on the test day was calculated by dividing the escape latency (s) achieved on the test day by that achieved on the first day.

### 2.4. Physical Growth Assessment

Dual-energy X-ray absorptiometry (DEXA; Hologic Discovery A, Bedford, MA, USA) was used to conduct a whole-body scan to measure tail length, bone mineral density (BMD; g/cm^2^), bone mineral content (BMC; g), and body mass (g).

### 2.5. Plasma Biochemical Assay

Plasma glucose levels were measured using a dry-chemistry blood analyzer, Spotchem SP-4410 (Arklay, Kyoto, Japan), according to the manufacturer’s instructions. Plasma insulin levels were measured using an enzyme-linked immunosorbent assay (ELISA) kit (Millipore, St Charles, MO, USA). Briefly, 10 μL plasma samples (and a standard solution) were loaded onto a pre-coated plate, incubated (2 h, room temperature) with 80 μL of detection antibody, washed with wash buffer, incubated (30 min, room temperature) with 100 μL of enzyme solution, washed again, and finally incubated (10 min, room temperature) with 100 μL of substrate solution. Sample absorbance at 450 nm was read immediately after the addition of 100 μL stop solution.

### 2.6. Hippocampal Transcriptome Analysis

#### 2.6.1. RNA Extraction and Next-Generation RNA Sequencing (RNA-seq)

The total RNA was extracted from the hippocampus of each animal using a DNA-free RNA isolation kit (RNAqueous-4PCR kit; Ambion, Austin, TX, USA) as previously described [21], and then total hippocampal RNAs from each group were pooled. The total RNA integrity and quantity were assessed using an Agilent 2100 bioanalyzer (Agilent Technologies, Palo Alto, CA, USA), before sequencing libraries were prepared. Briefly, polyadenylated mRNA was isolated from 3 µg samples of total RNA using a MicroPoly(A)Purist kit (Thermo Fisher Scientific, Wilmington, DE, USA) according to the manufacturer’s instructions. The mRNA samples were then fragmented using RNase III from the Ion Total RNA-Seq Kit v2 (Thermo Fisher Scientific, Wilmington, DE, USA), and converted to cDNA using Ion Total RNA-Seq Kit v2. Barcoded cDNA libraries were prepared using the Ion Xpress™ RNA-Seq Barcode kit (Life Technologies, Carlsbad, CA, USA) and the Ion Xpress™ RNA3’Barcode Primer (Life Technologies, Carlsbad, CA, USA), and quantified using the Agilent 2100 bioanalyzer. cDNA fragments > 200 bp in length were selected for use, and the cDNA library was diluted to 100 pmol and subjected to emulsion PCR (em-PCR) on Ion Sphere Particles (ISPs) using the Ion PI™ OT2 200 Kit and the Ion OneTouch™ System (Life Technologies, Carlsbad, CA, USA) according the manufacturer’s instructions. Enriched template-positive ISPs were loaded onto the Ion Proton PI chip v3 (Life Technologies, Carlsbad, CA, USA). Next-generation RNA-seq of the enriched libraries was performed using the Ion Proton semiconductor sequencer (Thermo Fisher Scientific, Wilmington, DE, USA).

#### 2.6.2. Bioinformatic Analysis of RNA-seq Data 

Sequencing reads were mapped to the mouse reference genome (NCBI mm9) using the TopHat and Bowtie algorithms, and raw read counts were normalized to reads/kb of exon model/million mapped read (RPKM) values using Partek® Genomics Suite software v6.6 (Partek, St Louis, MI, USA; http://www.partek.com/partekgs), before being mapped to transcripts. Average gene expression levels were compared between groups, and calculated *p* values were corrected for multiple comparisons using a False Discovery Rate (FDR) algorithm. Transcripts with RPKM > 1.5 and ‘significant’ genes (fold change > 1.3 and <-1.3) were selected for further analysis. The biological functions of significant genes were categorized via an Ingenuity Pathway Analysis (IPA) [22]. Significant functional categories were determined via a right-tailed Fisher’s exact test. Mechanistic networks linking signaling and metabolic pathways were predicted using Ingenuity Knowledge Base software [22].

### 2.7. Quantitative Real-Time Polymerase Chain Reaction (qRT-PCR)

qRT-PCR was conducted to validate RNA-seq data describing brain neurotrophic factor expression. DNase I-treated total RNA (extracted from either the hippocampus, or whole brain excluding the hippocampal region) was converted into cDNA using the MessageSensor RT kit (Ambion, Austin, TX, USA). mRNA levels were quantified using the SYBR-GREEN qPCR system (Applied Biosystem, Carlsbad, CA, USA), and primers for glyceraldehyde-3-phosphate dehydrogenase (Gapdh; forward, 5’-TGCACCACCAACTGCTTAG-3’; reverse, 5’-GATGCAGGGATGATGTTC-3’), nerve growth factor (Ngf; forward, 5’-AGACTCCACTCACCCCGTG-3’; reverse, 5’-GGCTGTGGTCTTATCTCCAAC-3’), neurotrophin 3 (Nt-3; forward, 5’-GGAGTTTGCCGGAAGACTCTC-3’; reverse, 5’-GGGTGCTCTGGTAATTTTCCTTA-3’), and Bdnf (forward, 5’-AAAGTCCCGGTATCCAAAGGCCAA-3’; reverse, 5’-TAGTTCGGCATTGCGAGTTCCAGT-3’). Relative mRNA expression levels (i.e., to that of Gadph) were calculated using the ΔΔCT method. 

### 2.8. Statistical Analysis 

Comparisons between two groups were assessed using an unpaired Student’s *t*-test. Comparisons of behavior-test data collected on the first training and final test day were assessed using a paired Student’s *t*-test. A Pearson’s correlation analysis was performed to evaluate relationships between two variables. Significant functional gene categories were determined via a Fisher’s exact test, (*p*-values are provided with transcriptome analyses). The FDR algorithm was used to correct for multiple testing. *P* values < 0.05 and FDR values <0.05 were considered to indicate statistical significance. Statistical analyses were performed using GraphPad Prism 6 software (GraphPad Software Inc., La Jolla, CA, USA), Partek® Genomics Suite software v6.6 (Partek, St Louis, MI, USA), and/or IPA.

## 3. Results

### 3.1. Long-term Dehydration Induces Growth Retardation in Young Mice

We previously showed that mice subjected to the dehydration-inducing protocol utilized herein consume 60% less water than CON mice over the four week long course of the experimental period, and become significantly hyperosmolar, (DEH, 346 mOsm/kg; CON, 334 mOsm/kg; *p* < 0.05) [11]. Herein, we examined whether dehydration induces metabolic changes and/or growth retardation in young mice. No significant difference in energy intake was observed between the CON and DEH mice during the experimental period (Appendix A), and although we noted that plasma glucose levels were significantly (20%) lower (*p* < 0.05) in the DEH than the CON group, this change was not accompanied by any significant changes to plasma insulin levels (Appendix A). Interestingly, the conducted DEXA total body scan showed that despite exhibiting the same daily energy intake, mice in the DEH group showed reduced physical growth compared to those in the CON group (Table 1). Specifically, the DEH mice exhibited a 5% (*p* < 0.05) reduction in lean body mass and a 9% (*p* < 0.05) reduction of their tail length, which is considered to be a standard growth indicator of rodent growth [23]. Therefore, these data suggest that dehydration incurs deleterious impacts on physical growth during the examined developmental period in young mice.

### 3.2. Dehydration Markedly Impacts Spatial-Learning Behavior in Young Mice

To evaluate the effects of dehydration on cognitive development, we subjected mice to the Barnes maze test. Specifically, mice were allowed an adaptation day and then trained for four consecutive days during the last week of the experimental period. We observed significant differences in the escape latency that was exhibited and in the total distance that was moved by the CON versus the DEH mice to reach the target escape hole (Figure 1a,b). The CON mice showed a significant reduction in their primary escape latency (*p* < 0.001) and total distance moved (*p* < 0.05) over the course of the training phase, whereas in contrast, the DEH mice exhibited relatively small improvements in both values (Figure 1a,b). Consistent with this, we observed gradual improvements in the performance, including significant increases in the speed (*p* < 0.001) of the CON, but not the DEH mice completing the Barnes maze test (Figure 1c). By the final test day, the CON mice headed directly to the escape hole, while the DEH mice instead wandered around the maze before eventually finding the escape hole (Figure 1d). Finally, the CON mice exhibited a significantly (approximately 30%) reduced relative latency (*p* < 0.001) on the final test compared to the first training day, while in contrast, DEH mice tended to show an increased latency of up to 430% between the two time points (*p* = 0.34) (Figure 1e). Taken together, these data support that dehydration detrimentally impacts cognitive function in young mice.

### 3.3. Dehydration Alters Gene Networks Associated with Glucose Metabolism and LTP Signal Transduction

Next, we conducted next-generation RNA-seq to assess gene expression patterns in the hippocampal region of the brain. Pooled total hippocampal RNA from each group was sequenced, and bioinformatic analyses were performed to identify differentially regulated transcriptional profiles between the two groups as described in the Methods. A total of 3197 genes were identified as differentially expressed genes (DEGs) between the CON and DEH groups. Functional categorization of these DEGs showed that the most significantly affected functional categories included behavior, cell-to-cell signaling, and neurological disorders (Figure 2a). We also performed a transcriptional network analysis which revealed that the most significantly impacted network by dehydration was carbohydrate metabolism, consistent with our previous finding that DEH mice are hypoglycemic (Appendix A). This network included genes known to mediate glucose uptake (*Slc2a3, Slc45a1, Slc50a1*) or neurotransmitter-sodium symporter activity (*Slc6a17, Slc6a20a, Slc6a9*), as well as key regulatory glucose-metabolizing enzymes (*G6pc3, Hk1, Ldha, Pfkfb3*, *Pfkp*) (Figure 2b). 

In addition, transcriptional networks involved in synaptic organization and adhesion were differentially regulated in the DEH compared to the CON mice (Figure 2c); for example, transcripts of CREB1 and CREB-mediated downstream molecules known to mediate synaptic plasticity (*Egr1*, *Fos*, *JunB*, *Nr4a1*, *Arc*) or neuronal-cell growth and survival (*Gadd45b*, *Midn*, *Btg2, Bdnf*) were downregulated in response to long-term dehydration (Figure 2d). Finally, a pathway analysis was conducted to evaluate whether dehydration suppressed long-term potentiation (LTP) that is a critical process for spatial memory to produce the cognitive deficits observed in the DEH mice. Interestingly, transcript levels of genes involved in LTP signaling were significantly dysregulated (*p* < 0.05) (Figure 2e). These results highlight that sustained insufficient water intake alters the transcription of genes that regulate neurogenesis and neurotransmission, and thereby likely promotes cognitive dysfunction.

### 3.4. Dehydration Modulates Hippocampal Bdnf Expression 

Given that dehydration was shown to affect transcriptional networks related to neurogenesis, we next compared the brain weight of mice in the DEH and CON groups and conducted a qPCR analysis to assess the expression levels of neurotrophic factors including *Ngf*, *Nt-3*, and *Bdnf*. Notably, while neither the brain weights nor the expression of other neurotrophins did not differ (Figure 3a,b), hippocampal *Bdnf* expression was significantly (*p* < 0.05) and specifically decreased in the DEH compared to the CON group (Figure 3b), (no significant changes to *Bdnf* levels were detected in whole brain regions excluding the hippocampus) (Figure 3c). Moreover, hippocampal *Bdnf* transcript levels were significantly associated with plasma osmolality levels (R^2^ = 0,495; *p* < 0.005) (Figure 3d). Altogether, these results suggest that dehydration may induce hippocampal *Bdnf*-mediated effects on cognitive function, and they support that hippocampal *Bdnf* may be a key driver of dehydration-induced pediatric cognitive dysfunction.

## 4. Discussion

Water is a vital nutrient that acts as a building material, a reaction medium and reactant, and a carrier for nutrients and waste products; therefore, optimal hydration is essential to maintain many biological system functions [1,24]. Water requirements for adequate hydration differ with age; for example, infants and children exhibit a higher risk of becoming dehydrated than adults if not offered sufficient water because of their immature fluid-regulatory system, higher body-water composition, higher metabolic rate, and poor thirst response [24,25]. However, previous studies have failed to provide strong and consistent evidence of the impacts of dehydration during infancy and childhood, often because the methodologies used to assess the effects of dehydration have varied widely between studies. For example, previous studies generally either examined the acute effects of 24 and/or 48 h of dehydration using a complete water-deprivation model [26,27], or induced dehydration using heat- and exercise-exposure models [8,28,29]. Neither of these methodologies evaluate the effects of long-term dehydration induced by insufficient water intake; thus, the present study aimed to examine the impacts of prolonged insufficient water intake on the physical and cognitive development.

Infancy and childhood are critical periods of physical growth and cognitive development, for which optimal hydration is imperative [12]. In the present study, we showed that plasma glucose levels were decreased in the DEH mice. Given that glucose levels have been shown to be positively associated with body weight, fat mass, and lean mass in children [30], it is plausible that a shortage of fuel for normal metabolism may have contributed to the physical growth retardation observed in the DEH mice. Moreover, a previous study showed that an increase in plasma vasopressin levels after 48 h of complete water deprivation disrupted cerebrovascular circulation, and thereby negatively affected neural function, and altered blood flow to the somatosensory cortex in animals [26]. Consistent with this, gene transcripts related to energy metabolism were shown herein to be differentially regulated in the hippocampus of DEH compared to CON mice; thus, it is likely that improper nutritional delivery may have disrupted brain function in the DEH mice.

Nevertheless, recent studies have failed to reach a consensus on the detrimental effect of dehydration on cognitive function. A few studies have shown that dehydration stimulated neural activity and increased attention and alertness [3,31]; however, others have shown that it increased neurophysiological stress and induced neural dysfunction [25,26]. In addition, a microarray analysis of the murine brain tissues after 24 h of water deprivation presented that dehydration altered the levels of transcripts associated with neurotransmitter transport, including that of glutamate, taurine, GABA, and aspartate, and reduced neuronal cell volumes; however, that study did not assess the effects of prolonged dehydration, nor conduct any behavioral assessment [27]. Herein, we used the Barnes maze test to demonstrate that DEH mice show impaired learning abilities, and furthermore, we examined the mechanisms underlying this effect by conducting a next-generation RNA-seq-based hippocampal transcriptome profiling analysis. Interestingly, we identified significant alterations to the expression of genes associated with synapse organization. The LTP signaling pathway, including its target molecule *Ntf* [32,33,34], is known to regulate diverse processes, including neuronal development, synaptogenesis, and plasticity. Notably, *Bdnf*, which is a major element of LTP signaling, was herein shown to be downregulated in the hippocampus, but not any other brain regions in the DEH mice, and this change was not accompanied by any significant perturbations of other neurotrophic factors. It is plausible that localized changes to hippocampal *Bdnf* expression may inhibit LTP formation and maintenance; however, further functional studies are required to confirm whether and examine how long-term dehydration impacts hippocampal *Bdnf* expression to modulate cognitive function.

Thus, the present study provides novel and invaluable insights into the effects of long-term dehydration on the physical growth of young mice and data pertaining to its molecular effects on cognitive function. However, further studies using rehydration and/or dose–response dehydration models are needed to confirm the effects of dehydration on cognitive function and characterize hydration requirements in young mice. Moreover, extrapolation from animal studies to humans should be carefully considered in terms of age, dietary and learning behaviors, and other factors [35].

## 5. Conclusions

To the best of our knowledge, this is the first study to identify that sustained dehydration causes physical retardation, cognitive deficits, and an altered transcriptional network in LTP-Bdnf signaling in young mice. Our findings provide novel mechanistic evidence to support that optimal hydration is essential to achieve normal physical and cognitive development during infancy and childhood; thus, it highlights that caregivers should be sensitive to the signs of dehydration in infants and children to achieve their optimal physical growth and cognitive development.

## Figures and Tables

**Figure 1 nutrients-12-00670-f001:**
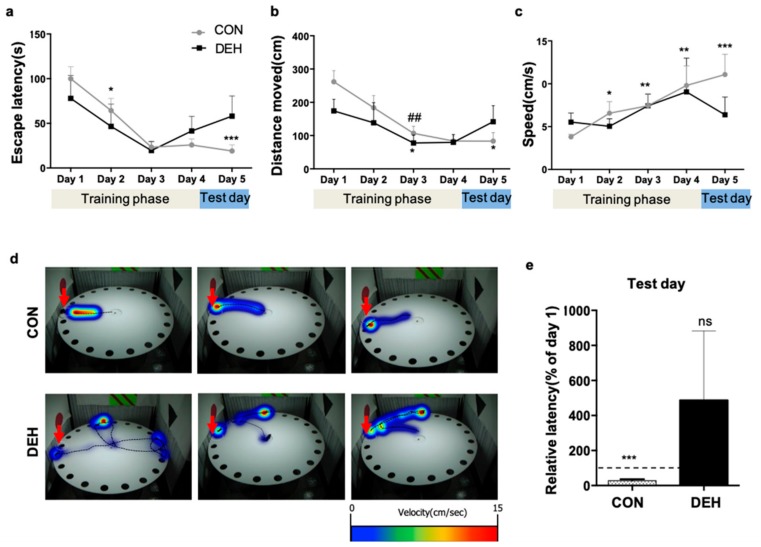
Dehydration markedly alters spatial learning behavior in young mice. Spatial learning behavior was measured in control (CON) and dehydrated (DEH) mice by using the Barnes maze test to measure the (**a**) primary escape latency, (**b**) total distance moved, and (**c**) speed (calculated by dividing total distance moved by the primary latency) during the 4-day maze ‘training’ phase. (**d**) Representative heatmap images (i.e., heatmap indicates time spent at each position) tracking the path taken by mice through the maze on the final ‘test’ day. Arrow, target escape hole. (**e**) The average relative latency (%) required to identify the escape hole on the test day was calculated by dividing the escape latency (s) achieved on the test day by that achieved on the first day. Data are presented as the mean ± SEM. Statistical significance was evaluated using a Student’s *t*-test: * *p* < 0.05, ** *p* < 0.005, *** *p* < 0.001 versus day 1 in the DEH group; ## *p* < 0.01 versus day 1 in the CON group. NS, not statistically significant.

**Figure 2 nutrients-12-00670-f002:**
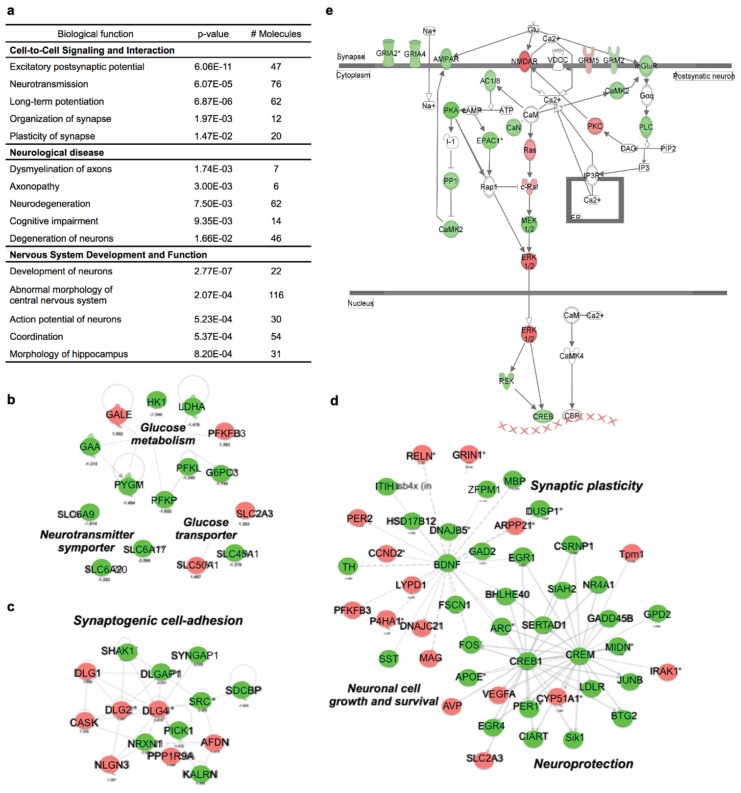
Dehydration alters hippocampal transcriptional networks. A hippocampal transcriptional network analysis was performed, comprising next-generation RNA sequencing and bioinformatic analyses. (**a**) Genes that were differentially regulated between the control (CON) and dehydrated (DEH) mice were classified into functional categories via an Ingenuity Pathway Analysis (IPA). Significantly changed biological functions are indicated by *p*-values calculated using a Fisher’s exact test, and the number of distinctively modulated molecules in each functional category is indicated. (**b**–**d**) Transcriptional networks related to altered glucose metabolism, synaptogenic cell-adhesion, and synaptic plasticity were differentially regulated in response to long-term dehydration. (**e**) Genes associated with long-term potentiation (LTP) signaling transduction were differentially up- (red) or downregulated (green) in the DEH compared to the CON mice.

**Figure 3 nutrients-12-00670-f003:**
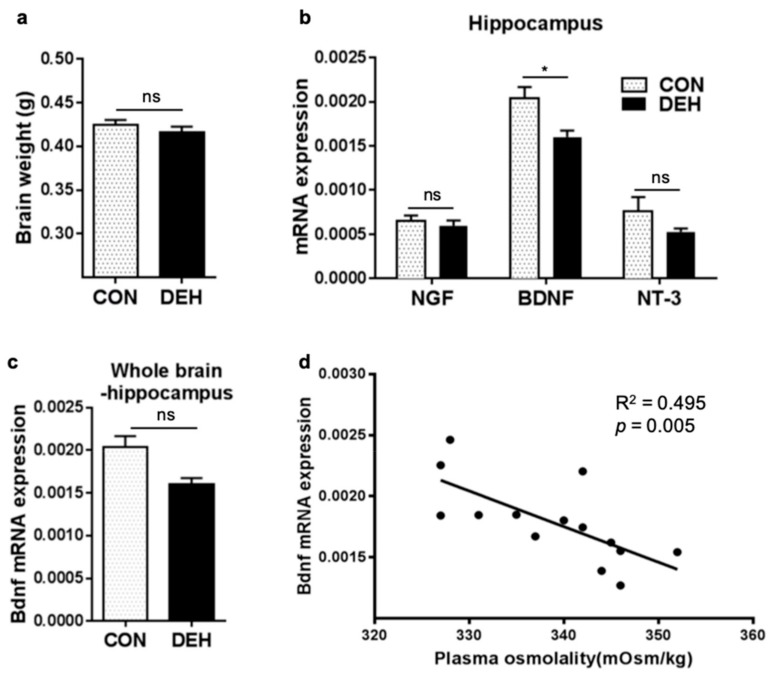
Dehydration alters hippocampal brain-derived neurotrophic factor (Bdnf) expression levels in young mice. (**a**) Brain weight was measured after the dehydration experiment in the control (CON) and dehydration group (DEH). (**b**) The expression of neurotrophic factors in the hippocampus and (**c**) of Bdnf mRNA in other brain regions (excluding the hippocampus) were assessed via qPCR. (**d**) The relationship between hippocampal Bdnf transcript and plasma osmolality levels was evaluated via a Pearson’s correlation analysis. Data are presented as the mean ± SEM. * *p* < 0.05 versus CON group according to a Student’s *t*-test. R^2^ and p-values were calculated via a Pearson’s correlation analysis. NS, not statistically significant.

**Table 1 nutrients-12-00670-t001:** Dehydration induces physical growth retardation in young mice.

	CON	DEH	*p* Value
Lean mass, g	20.89 ± 0.26	19.07 ± 0.63	0.03
Fat, %	13.93 ± 1.09	9.98 ± 0.38	0.06
Area, cm^2^	3.90 ± 0.31	3.59 ± 0.48	0.57
BMC, g	0.44 ± 0.04	0.38 ± 0.04	0.32
BMD, g/cm^2^	0.11 ± 0.01	0.11 ± 0.01	0.22
Relative tail length	22.63 ± 0.77	20.16 ± 0.38	0.04

Physical growth was measured with body composition, bone mineral content (BMC), bone mineral density (BMD), and tail length using dual-energy X-ray absorptiometry (DEXA) in the control (CON) and dehydration (DEH) groups. Statistical significance was evaluated by Student’s t-test between two groups. Data are presented as the mean ± SEM.

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
