# Peer review of "Dehydration Impairs Physical Growth and Cognitive Development in Young Mice"

_nutrients, 2020, doi:10.3390/nu12030670_

Round 1
Reviewer 1 Report
Dear Authors,
I commend you on the well designed and written scientific paper.
The abstract is well written and concise with key words provided.
The rationale for the study was given in the introduction, with broad and balanced research as background. The link between dehydration and cognition is under researched, so the use of controlled experimentation is great.
You have also looked at the same concept using different approaches, which is great.
The method is detailed and authors have used standard methods.
Authors need to be cautious in extrapolating the results of this study directly to humans. You can have a look at the reference AKHTAR (2015) The Flaws and Human Harms of Animal Experimentation. Cambridge Quarterly of Healthcare Ethics 24: 407–419 .
This should be discussed as part of the limitations of the study and possibly one or two sentences about future plans will be great.
Overall, a well written paper.
Author Response
Response to Reviewer 1 Comments
Point 1: Authors need to be cautious in extrapolating the results of this study directly to humans. You can have a look at the reference AKHTAR (2015). The Flaws and Human Harms of Animal Experimentation. Cambridge Quarterly of Healthcare Ethics 24: 407–419.
This should be discussed as part of the limitations of the study and possibly one or two sentences about future plans will be great.
Response 1: We appreciate for this thoughtful suggestion. As suggested, we discussed this in the revised manuscript (L 344-346).
Reviewer 2 Report
Using a mouse model (three-week-old C57BL/6J mice) the authors studied effects of sustained dehydration on certain cognitive development parameters by means of the Barnes maze test. They report that dehydrated mice exhibited a reduced body mass and tail length, and spent four times longer completing the Barnes maze test than control mice. Also, dehydration significantly dysregulated LTP signaling and specifically decreased hippocampal BDNF mRNA expression levels.
Comments
- 1. Fig. 1E shows that in DEH mice escape latency on the test day tends to be higher, albeit not significantly, compared with escape latency in DEH mice on day 1. However, Fig. 1A shows that escape latency in DEH mice on test day amounts to ca. 60 s as opposed to ca. 80 s on day 1, which would lead to a ca. 25% decrease in the test day/day1 ratio. The authors should explain this discrepancy.
- 3B shows that sustained dehydration causes a modest, though significant decrease in hippocampal BDNF mRNA expression levels while leaving essentially unaltered hippocampal NGF and NT-3 mRNA expression levels. The authors should show that the alteration of BDNF mRNA levels translates in decreased levels of BDNF protein while leaving unchanged levels of NGF and NT-3. Relying on mRNA levels might not be enough to conclude that dehydration specifically reduces BDNF.
Author Response
Response to Reviewer 2 Comments
Point 1: Fig. 1E shows that in DEH mice escape latency on the test day tends to be higher, albeit not significantly, compared with escape latency in DEH mice on day 1. However, Fig. 1A shows that escape latency in DEH mice on test day amounts to ca. 60 s as opposed to ca. 80 s on day 1, which would lead to a ca. 25% decrease in the test day/day1 ratio. The authors should explain this discrepancy.
Response 1: We apologize for the confusion. Fig 1A depicts escape latency (sec) and Fig 1E shows relative latency (%). The relative latency (%) required to identify the escape hole on the test day was normalized by dividing the escape latency (sec) achieved on the test day by that achieved on the first day in each mouse, which means that the relative latency on the first day was 100% in each mouse. For example, one mouse in the DEH group spent 4 sec on day 1 and 130s on day 5 (3250% change on the test day compared to 100% on day 1). We described detailed method to measure the escape latency (sec) and the relative latency (%) in the Method section (L105-109).
Point 2: 3B shows that sustained dehydration causes a modest, though significant decrease in hippocampal BDNF mRNA expression levels while leaving essentially unaltered hippocampal NGF and NT-3 mRNA expression levels. The authors should show that the alteration of BDNF mRNA levels translates in decreased levels of BDNF protein while leaving unchanged levels of NGF and NT-3. Relying on mRNA levels might not be enough to conclude that dehydration specifically reduces BDNF.
Response 2: We agree with the reviewer, however, please understand that the size of hippocampus in a mouse is not large enough to determine both levels of mRNA and protein. In addition to qPCR assay for the gene, we performed RNA-sequencing analysis, which showed that the transcriptional networks of LTP signaling and synaptic plasticity known to be orchestrated by BDNF, are significantly downregulated in the DEH group. We believe this observation that downstream targets of BDNF were dramatically decreased in their expressions might support the reduced activity of BDNF in dehydrated mice.
Round 2
Reviewer 2 Report
The authors have satisfied my previous criticism.